# Multi-photon Rabi oscillations in the presence of the classical noise in a quantum nonlinear oscillator

Bogdan Y. Nikitchuk[1], Evgeny V. Anikin[1], Natalya S. Maslova[2]

**1** Russian Quantum Center, Skolkovo, 143025 Moscow, Russia
**2** Quantum Technology Centrum, Department of Physics, Lomonosov Moscow State University, 119991 Moscow, Russia

## Abstract

We consider the model of a single-mode quantum nonlinear oscillator with the fourth (Kerr) and sixth (over-Kerr) orders of nonlinearity in the presence of fluctuations of the driving field. We demonstrate that the presence of the amplitude noise does not significantly affect the multi-photon Rabi transitions for the Kerr oscillator, and, in contrast, suppresses these oscillations for the over-Kerr oscillator. We explain the suppression of multi-photon transitions in the over-Kerr oscillator by quasienergy fluctuations caused by noise in field amplitude. In contrast, for the Kerr oscillator, these fluctuations cancel each other for two resonant levels due to the symmetry in the oscillator quasienergy spectrum.

# 1 Introduction

The model of the quantum anharmonic oscillator is ubiquitous in nanoscale physics. It describes many systems important for modern applications of quantum technologies, including superconducting nanostructures [1] and qubits [2, 3], nanomechanical systems [4], and cold trapped-ions and atoms [5]. Recent technological advances allow for such systems to exhibit prominent nonlinearity on the few-quantum level. This opens a route to new approaches for controlling the state of quantum matter, in particular, the preparation of non-classical states of light [6, 7, 8, 9, 10]. Furthermore, nonlinear oscillator networks were suggested for universal quantum computation [11, 12]. Also, from the fundamental point of view, the interplay of quantum effects with nonlinearity gives rise to new fascinating phenomena including dissipative phase transitions [13] and dynamical tunnelling [14].

One of the intriguing phenomena which anharmonic oscillator can exhibit in the ultra-quantum regime is multi-photon transitions. For decades, multi-photon transitions have been the focus of active research in various systems. Moreover, they have been proposed as a tool to manipulate the quantum state of a number of nanoscale systems, for example, to control spin-mixing dynamics in a gas of spinor atoms [15] or in application to quantum gates in silicon-vacancy centres of SiC [16]. Also, multi-photon transitions can be useful for continuous wave electron paramagnetic resonance spectroscopy [17]. Thus, a deeper understanding of multi-photon transitions in quantum nonlinear oscillator will facilitate new approaches to control the quantum state of related systems.

In the anharmonic oscillator driven by a weak field, multi-photon transitions can occur between its eigenstates approximated by the Fock states [18, 19, 20, 21, 22]. For that, the driving field frequency should be detuned from the oscillator frequency to precisely compensate for the nonlinear frequency shift between the states. In this work, we analyse the effect of driving field fluctuations on multi-photon transitions in quantum nonlinear oscillator.

Typically, the value of multi-photon transition amplitude is quite small, which results in a narrow multi-photon transition width. Because of that, one should expect high sensitivity of multi-photon transitions to the driving field fluctuations. However, it was demonstrated previously that multi-photon transition frequencies in the model of the oscillator with Kerr nonlinearity are independent of the driving field amplitude due to the special symmetry of the model [18, 21, 22]. With numerical simulations and analytical arguments, we show that this property (which also manifests as the symmetry of the perturbation theory corrections) leads to the surprising robustness of the multi-photon transitions of the Kerr oscillator to field amplitude fluctuations.

Also, we analyse the model of the Kerr oscillator with additional high-order nonlinearity. It was shown that high-order nonlinearity breaks the symmetry of the perturbation theory corrections and leads to the amplitude-dependent shift in positions of multi-photon resonances [23]. With the help of the two-level effective model for two resonant oscillator levels, we prove that the presence of such a shift strongly increases the sensitivity of multi-photon transitions to amplitude fluctuations.

Our results provide the necessary conditions for experimental observation of multi-photon Rabi oscillations. We believe that our results enrich the tool-kit for quantum state manipulation of the nonlinear oscillator.

## 2   The model. Multi-photon Rabi oscillations

In this manuscript, we study multi-photon transitions in the model of a weakly nonlinear oscillator in rotating-wave approximation driven by the resonant driving field. We write the model Hamiltonian as

$$H_0 = \omega a^\dagger a + \frac{\alpha}{2} \left( a^\dagger a \right)^2 + \kappa \left( a^\dagger a \right)^3 + G(t) a^\dagger + G^*(t) a, \tag{1}$$

where $\omega$ is the oscillator frequency, $\alpha$ is the Kerr coefficient, $\kappa$ is the coefficient corresponding to the sixth-order nonlinearity, and $G(t)$ is the driving field. For non-zero values of $\kappa$ we reference the model (1) as an over-Kerr oscillator.

We explore weak deviations of the model Hamiltonian (1) from the pure Kerr Hamiltonian ($\kappa = 0$), so we don't take into account the powers of $(a^\dagger a)$ higher than 3 and consider only small values of $\kappa$. Also, we write the noisy driving field as

$$G(t) = g(t) \exp \left[ -i \int_0^t \Omega \left( t' \right) dt' \right], \tag{2}$$

where $g(t)$ and $\Omega(t)$ are the amplitude and the frequency of the driving field. After the unitary transformation with $U = \exp\left(-ia^\dagger a \int \Omega(t) dt\right)$, the Hamiltonian (1) transforms to

$$H = -\Delta(t) a^\dagger a + \frac{\alpha}{2} \left( a^\dagger a \right)^2 + \kappa \left( a^\dagger a \right)^3 + g(t) \left( a + a^\dagger \right), \tag{3}$$

where $\Delta(t) = \omega - \Omega(t)$ is the detuning of the driving field from the oscillator frequency.

At constant $g$ and $\Delta$, the oscillator can exhibit multi-photon transitions between some pair of states $|n\rangle$, $|n'\rangle$ providing that $\Delta$ is tuned to corresponding resonance. The resonant condition can be found easily for infinitely small $g$. In this case, the eigenstates of (3) are almost Fock states, and multi-photon transitions between the states $|n\rangle$ and $|n'\rangle$ occur (see Fig. 1) when $\epsilon_n^{(0)} = \epsilon_{n'}^{(0)}$, where $\epsilon_n^{(0)}$ is the quasienergy of the Hamiltonian (3) with $g = 0$. This is satisfied at the resonant value of the detuning which reads

$$\Delta_{\text{res}}^{(0)} = \frac{\alpha}{2} \left( n + n' \right) + \kappa \left( n^2 + nn' + n'^2 \right). \tag{4}$$

For the case of small but finite field amplitude $g$, each eigenstate becomes a superposition of Fock states. It still has a dominant contribution of the state $|n\rangle$ while the contributions of other Fock states are small in $g$. Therefore, resonant detuning becomes a function of $g$: $\Delta_{\text{res}} = \Delta_{\text{res}}(g)$. It can be found as a solution of the equation

$$\epsilon_n(g, \Delta_{\text{res}}) = \epsilon_{n'}(g, \Delta_{\text{res}}), \tag{5}$$

where $\epsilon_n(g, \Delta)$ are the energies of the oscillator eigenstates adiabatically evolving from $|n\rangle$. The subtleties of the definition of $\epsilon_n(g, \Delta)$ in the presence of degeneracy can be resolved by considering the energies as perturbation theory series [22]:

$$\epsilon_n = \epsilon_n^{(0)} + \sum_{k=1}^{\infty} |g|^{2k} \epsilon_n^{(2k)}, \tag{6}$$

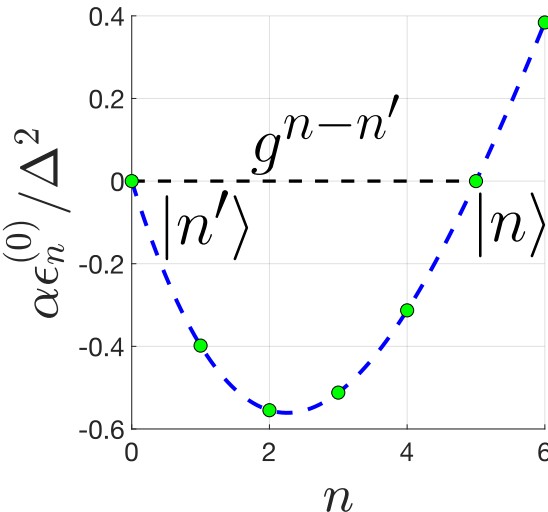

Figure 1: Dimensionless quasienergies $\alpha\epsilon_n^{(0)}/\Delta^2$ (for $g = 0$) as a function of Fock state number $n$. The detuning $\Delta$ is chosen so that the states $n = 5$, and $n' = 0$ are in resonance (their quasienergies are equal). The plot is shown for $\kappa/\alpha = -0.025$, and $\Delta/\alpha = 1.8750$.

here $\epsilon_n^{(k)}$ is the $k$-th order of non-degenerate perturbation theory correction to the quasienergy. Using Eqs. (4) and (6), it is straightforward to find $\Delta_{\text{res}}(g)$ as a perturbation theory series. Up to second order in $g$,

$$\Delta_{\text{res}}(g) = \Delta_{\text{res}}^{(0)} + \frac{\epsilon_n^{(2)} - \epsilon_{n'}^{(2)}}{n - n'}g^2 + o\left(g^2\right). \tag{7}$$

For pure Kerr oscillator, the following relation for the perturbation theory corrections is valid: $\epsilon_n^{(k)} = \epsilon_{m-n}^{(k)}$, $n = 0, \ldots, m$, and $k = 0, \ldots, m-2n$, where $m = 2\Delta_{\text{res}}^{(0)}/\alpha$ is an integer [18, 21, 22]. Because of that, the resonant detuning for transitions between levels $n$ and $n'$ does not depend on $g$ up to the order of $n - n'$. In contrast, $\epsilon_n^{(2)} \neq \epsilon_{n'}^{(2)}$ when $\kappa \neq 0$ (see Appendix A), which leads to a non-zero correction to $\Delta_{\text{res}}$ according to Eq. (7).

Because of the different dependence of the resonant detuning $\Delta_{\text{res}}$ on $g$, the quasienergies have different dependence on $g$ for the Kerr and over-Kerr models when $\Delta$ is near resonance (see Fig. 2). For the over-Kerr model, quasienergies of the states $n$ and $n'$ have an avoided crossing near the resonant value of $g = g_{\text{res}}(\Delta)$ determined by condition $\Delta_{\text{res}}(g_{\text{res}}) = \Delta$. The gap at $g_{\text{res}}$ is determined by multi-photon transition amplitude $\propto g_{\text{res}}^{n-n'}$. In contrast, for the Kerr model at $\Delta_{\text{res}} = \alpha(n + n')/2$, quasienergies of two states closely follow each other and differ by $\sim g^{n-n'}$ in a whole range of $g$.

Now, let us consider the driving field with time-dependent fluctuations which are always present in real systems. Due to the mentioned difference in the quasienergy dependence on $g$ for the Kerr and over-Kerr models, one can expect that the fluctuations in $g$ affect the multi-photon transitions differently in these models. More precisely, we show in Sections 3 and 4 that the driving field amplitude fluctuations do not affect the multi-photon Rabi transitions for the Kerr model. Roughly, this is explained by the fact that $\Delta_{\text{res}}$ does not depend on $g$, and small fluctuations in $g$ cannot lead the system out of the resonance. In contrast, for the over-Kerr model, we show that amplitude fluctuations suppress multi-photon transitions.

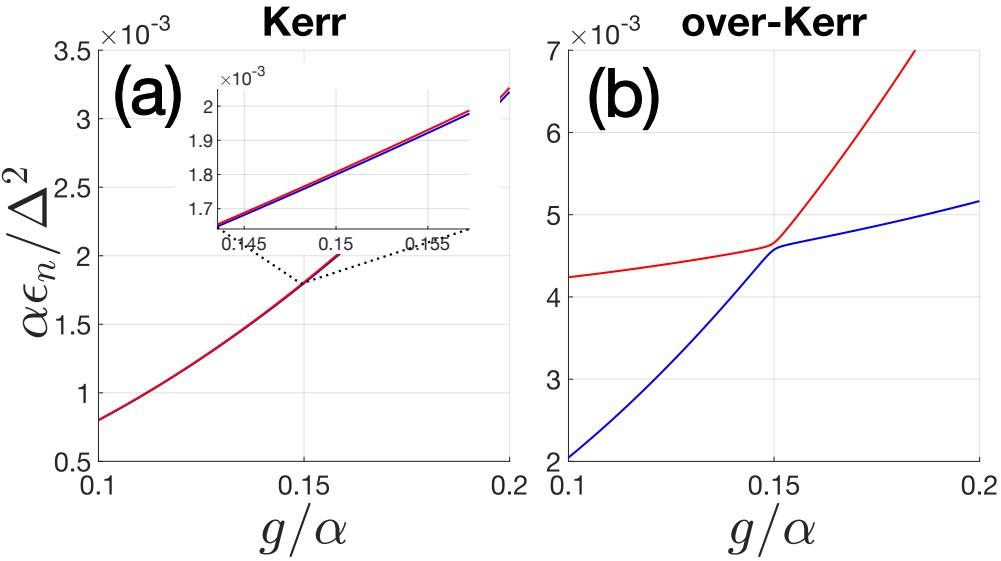

Figure 2: Dimensionless quasienergies $\epsilon_n \alpha / \Delta^2$ obtained by numerical diagonalization of the Hamiltonian (3) corresponding to two resonant states with $n = 5$, and $n' = 0$ as a function of $g$. The parameters are as follows: $\kappa/\alpha = 0$, $\Delta/\alpha = 2.5$ for (a), and $\kappa/\alpha = -0.025$, $\Delta/\alpha = 1.872$ for (b).

In Sections 3 and 4, we consider the Kerr and over-Kerr models in the presence of classical noise with finite correlation time and study the multi-photon Rabi oscillations. We model the noisy driving field by considering the time-dependent $g(t)$ and $\Delta(t)$ in the form $g(t) = g_0 + \xi_1(t)$, and $\Delta(t) = \Delta_0 + \xi_2(t)$, where $g_0$, and $\Delta_0$ — mean field amplitude and detuning respectively, and $\xi_{1,2}(t)$ are the amplitude and frequency fluctuations. We model them as real Gaussian processes with zero average and correlation functions

$$\langle \xi_i(t)\xi_j(t')\rangle = C_j\left((t-t')/\tau_j\right)\delta_{ij}, \quad i, j = 1, 2. \tag{8}$$

Here $\tau_i$ are correlation times, $\delta_{ij}$ is the Kronecker delta symbol, and the correlation functions $C_j(x)$ are assumed to decay away from zero at $|x| \sim 1$. Also, we denote the noise dispersion as $\sigma_j^2 \equiv C_j(0)$.

For the numerical simulations in Section 4 and part of the theoretical estimates in Section 3, we use the model of the exponentially correlated noise with

$$\langle \xi_i(t)\xi_j(t')\rangle = \sigma_j^2 \exp\left(-\frac{|t-t'|}{\tau_j}\right). \tag{9}$$

## 3 Two-level effective model

In this section, we consider the effect of driving field fluctuations on multi-photon Rabi oscillations between two resonant levels $n$, and $n'$ of the nonlinear oscillator.

We focus on the case of fluctuations with relatively large correlation time such as they do not cause direct transitions to non-resonant levels. In this case, it is possible to utilise a two-level effective Hamiltonian which takes into account the multi-photon transition amplitude

$\omega^R_{n,n'}$, the second-order corrections to the quasienergies of the levels $n$, $n'$, and their dependence on the fluctuating driving field amplitude $g(t)$ and detuning $\Delta(t)$

$$H_{\text{eff}} = \begin{bmatrix} \epsilon^{(0)}_{n'} + \epsilon^{(2)}_{n'} g(t)^2 - \xi_2(t)n' & \omega^R_{n,n'} \\ \omega^R_{n,n'} & \epsilon^{(0)}_n + \epsilon^{(2)}_n g(t)^2 - \xi_2(t)n \end{bmatrix}. \tag{10}$$

Here $\epsilon^{(0)}_n, \epsilon^{(2)}_n$ are defined in Section 2 and Appendix A, and the multi-photon Rabi frequency can be calculated as [24, 21]

$$\omega^R_{n,n'} = g^{n-n'} \sqrt{\frac{n!}{n'!}} \prod_{k=n'+1}^{n-1} \left( \epsilon^{(0)}_n - \epsilon^{(0)}_k \right)^{-1}. \tag{11}$$

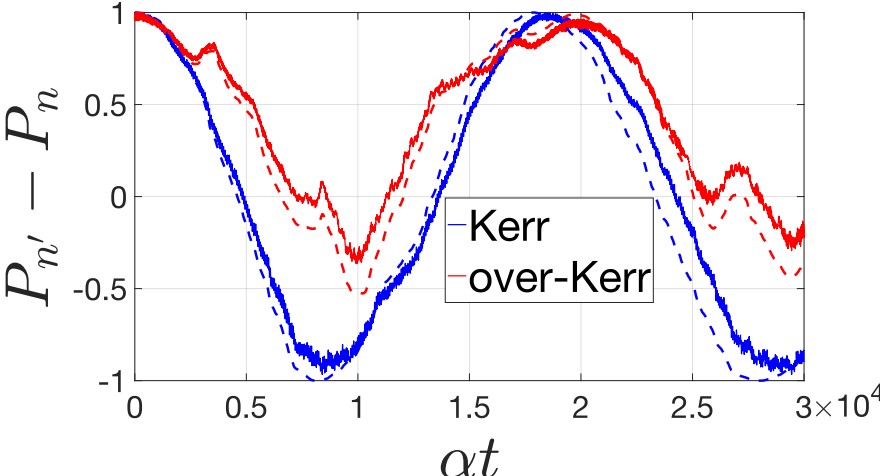

Figure 3: Comparison between the full model (3) (solid lines) and the effective (10) one (dashed lines) of the population inversions as a function of time for the Kerr and over-Kerr models. The parameters are: $n = 5$, $n' = 0$, $g/\alpha = 0.22$, $\Delta_{\text{full}}/\alpha = 2.5$, and $\Delta_{\text{two-level}}/\alpha = 2.5$ for Kerr; $g_0/\alpha = 0.15$, $\Delta_{\text{full}}/\alpha \approx 1.872229$, $\Delta_{\text{two-level}}/\alpha \approx 1.872223$, and $\kappa/\alpha = -0.025$ for over-Kerr. Parameters of the noise are: $\tau\alpha = 100$, $\sigma/\alpha = 0.022$ (Kerr), and $\sigma/\alpha = 0.015$ (over-Kerr).

For the applicability of the two-level model (10), the characteristic escape time from two resonant levels $n$ and $n'$ must be much larger than the period of the multi-photon Rabi oscillations. For example, for the exponentially correlated noise with the correlation function (9) the condition sufficient for that reads

$$\tau \gg 4\pi \frac{\eta^2 g_0^2}{\alpha^2 \omega^R_{n,n'}}, \tag{12}$$

where $\eta = \sigma/g_0$ (see Appendix B).

To justify further results, in Fig. 3 we compare the results of the numerical simulation of TDSE for the full (3) and effective (10) models for a single noise realisation. We consider a multi-photon transition by 5 quanta for a single realisation of exponentially correlated noise with correlation time $\tau$ such that the applicability condition (12) is valid. We plot

the population inversion $P_{n'} - P_n$ ($P_k$ is a probability to find the system in the state with quantum number $k$) as a function of time for both models and find good agreement between the two-level model full Hamiltonian evolution.

The resonant values of the detuning are slightly different for the full and effective models because only second-order correction to energy is taken into account in the effective model.

With the effective two-level model, we study the influence of noise on multi-photon transitions by averaging over the noise realizations. It is possible to find analytically the time evolution of the noise-averaged density matrix for the effective model $\rho = \langle\langle|\psi\rangle\langle\psi|\rangle\rangle$ (where $\langle\langle\ldots\rangle\rangle$ denotes averaging over noise realizations). For that, one should consider the noise correlation time to be much smaller than the characteristic time-scale of the noise-free effective Hamiltonian: $\tau \ll 2\pi/\omega_{n,n'}^R$, and the amplitude of the noise to be small in comparison with $g_0$.

This allows us to approximate the noise correlation functions with delta-functions: $\langle\xi_i(t)\xi_j(t')\rangle \approx Q_i\delta(t - t')\delta_{ij}$, where $Q_i = 2\tau_i\sigma_i^2$. Also, small noise magnitude allows the expansion of the Hamiltonian in first order in the noise amplitude

$$H(t) = H_0 + \sum_{j=1}^{2} \xi_j(t)V_j(t), \tag{13}$$

where $V_1$ corresponds to the amplitude noise and $V_2$ to the frequency noise. The operators in (13) read

$$H_0 = \begin{bmatrix} \epsilon_{n'}^{(0)} + \epsilon_{n'}^{(2)}g_0^2 & \omega_{n,n'}g_0^{n-n'} \\ \omega_{n,n'}g_0^{n-n'} & \epsilon_n^{(0)} + \epsilon_n^{(2)}g_0^2 \end{bmatrix},$$

$$V_1 = \begin{bmatrix} 2\epsilon_{n'}^{(2)}g_0 & \omega_{n,n'}(n-n')g_0^{n-n'-1} \\ \omega_{n,n'}(n-n')g_0^{n-n'-1} & 2\epsilon_n^{(2)}g_0 \end{bmatrix}, \quad V_2 = \begin{bmatrix} -n' & 0 \\ 0 & -n \end{bmatrix}. \tag{14}$$

For the TDSE with the Hamiltonian Eq. (13), the noise-averaged density matrix obeys the master equation [25, 26]:

$$\dot{\rho} = -i[H_0, \rho] + \sum_{j=1}^{2} \frac{Q_j}{2}\left(2V_j\rho V_j - \{V_j^2, \rho\}\right). \tag{15}$$

In Appendix C, we discuss the derivation of Eq. (15) and obtain the analytical solution. The operator $V_1$ in Eq. (14) contains diagonal terms $V_{11}$, and $V_{22}$ responsible for the fluctuations of the energies of the Fock states and non-diagonal terms $V_{12}$ responsible for the fluctuations of the multi-photon amplitude. For the over-Kerr model, one can neglect the off-diagonal terms because $V_{12} \ll \min(V_{11}, V_{22})$. In this case, the time dependence of the population inversion $\sim \langle\sigma_z(t)\rangle = \text{Tr}(\rho(t)\sigma_z)$ reads (see Appendix C)

$$\langle\sigma_z(t)\rangle = e^{-\Gamma t}\left[\cosh\left(t\sqrt{\mathcal{D}}\right) + \frac{\Gamma}{\sqrt{\mathcal{D}}}\sinh\left(t\sqrt{\mathcal{D}}\right)\right], \tag{16}$$

where

$$\Gamma = Q_1g_0^2\left(\epsilon_n^{(2)} - \epsilon_{n'}^{(2)}\right)^2 + Q_2(n-n')^2/4,$$

$$\mathcal{D} = \Gamma^2 - 4\left(\omega_{n,n'}^R\right)^2. \tag{17}$$

As one can see from Eq. (17), the contribution of the frequency noise to the decay rate is small iff $Q_2 \ll \omega_{n,n'}^R$. Otherwise, the frequency noise significantly suppresses the Rabi oscillations in both models. Now, let us focus on the amplitude noise only ($Q_2 = 0$) because it leads to significantly different behaviour for the Kerr and over-Kerr models.

In the over-Kerr model, the decay rate is given by Eq. (17). In contrast, as Eq. (17) gives zero value of the decay rate for the pure Kerr model, one should not neglect non-diagonal terms in the operator $V_1$ in this case. However, due to a simple form of the Schroedinger equation (13) for pure Kerr case, it is possible to obtain the decay rate for $\sigma_z$ for arbitrary correlated Gaussian noise (see Appendix C):

$$\langle \sigma_z(t) \rangle \;=\; \exp\left[-2V_{12}^2 \int\limits_{[0,t]^2} \langle \xi(t')\xi(t'')\rangle dt'dt''\right] \cos\left(2\omega_{n,n'}^R t\right) \;\to\; e^{-2Q_1 t V_{12}^2} \cos\left(2\omega_{n,n'}^R t\right). \quad (18)$$

The last equality indicated by the arrow corresponds to the limit of delta–correlated noise.

The solutions (16) and (18) differ both quantitatively and qualitatively. First, the decay rate $\Gamma$ for the over-Kerr model is typically larger than that of the pure Kerr model. Second, $\langle \sigma_z(t) \rangle$ has different time dependencies in these cases. For the pure Kerr model, the $\langle \sigma_z(t) \rangle$ shows damped oscillations for all $Q_1$ and quickly approaches zero at large $Q_1$. This indicates that multi-photon oscillations only acquire random phase because of the fluctuations of multi-photon transition amplitude but are not destroyed by them. In contrast, the behaviour of $\langle \sigma_z(t) \rangle$ governed by Eq. (16) depends on the sign of $\mathcal{D}$. For weak enough noise (underdamped regime), $\mathcal{D} < 0$, and $\langle \sigma_z(t) \rangle$ shows damped oscillations like for the pure Kerr case. However, for larger values of noise (critically damped and overdamped regimes), $\mathcal{D} \geqslant 0$, and $\langle \sigma_z(t) \rangle$ shows slow monotonic decay. Therefore, strong enough noise significantly suppresses multi-photon transitions in the over-Kerr oscillator.

Let us examine at which conditions the system reaches the overdamped regime within our approximations. For that, it is necessary that $\Gamma > 2\omega_{n,n'}^R$. Also, for the applicability of the white-noise approximation, it is necessary $\tau \ll (\omega_{n,n'}^R)^{-1}$. For both of these conditions to be satisfied, $g_0$ should be small enough (with a correspondingly large multi-photon period). In Appendix D, we find the upper bound on $g_0$ and lower bound $\tilde{T}$ on the multi-photon period necessary to reach the overdamped regime.

We found out that $\tilde{T}$ scales as $|\kappa|^{-1-2/(n-n'-2)}$ at small $\kappa$. This indicates that the sensitivity of the multi-photon transitions to the amplitude noise increases with increasing $n - n'$. Indeed, at a lower value of $\tilde{T}$, weaker noise leads the system to the overdamped regime and thus completely suppresses multi-photon Rabi oscillations. The calculated lower bound $\tilde{T}$ is shown in Fig. 4 as a function of $\kappa$ for transitions by 3, 4, and 5 quanta. One can see that $\tilde{T}$ for the transition by 3 quanta is considerably larger than for the transition by 4 and 5 quanta.

## 4 Numerical results

Now, let us confirm the analytical results by the numerical simulations. For the Kerr and over-Kerr models, we compare the analytical results of section 3 with the results of the numerical TDSE solution for the full model and the effective two-level model. We chose different model parameters (see Table 1) to satisfy resonance conditions for different pairs of states $|n\rangle$, $|n'\rangle$, and found the population inversion between the resonant states as a function of time. For the

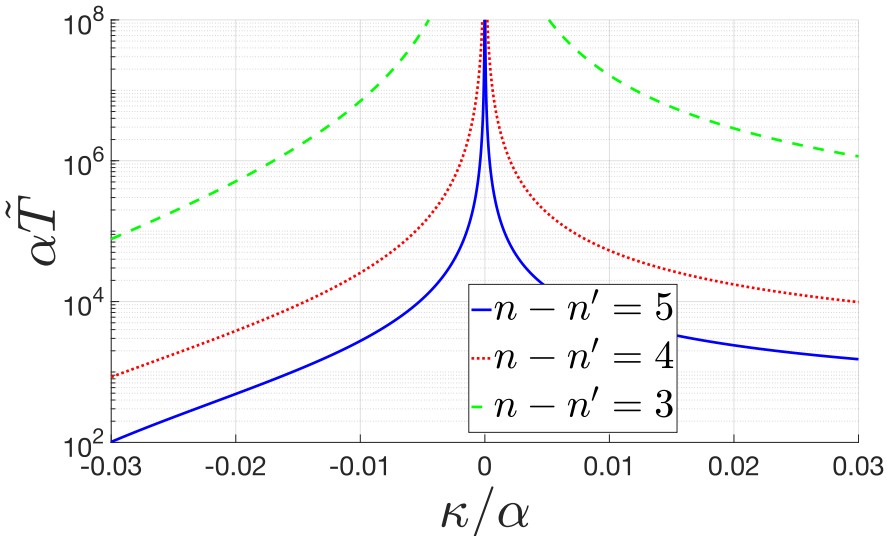

Figure 4: The lower bound $\tilde{T}(\kappa)$ for the period of multi-photon oscillations at which the system reaches the overdamped regime as a function of $\kappa$ for different $n - n'$. Here $\eta = 0.1$, and $\Delta = \Delta_{\text{res}}^{(0)}$.

noise model, we considered pure amplitude field fluctuations with the exponentially correlated random term $\xi_1(t)$ (see Eq. (9)). The Fock state $|n'\rangle$ has been chosen as an initial one.

| № | $n'$ | $n$ | $\kappa/\alpha$ | $g/\alpha$ | $\alpha\tau$ | $\Delta_{\text{full}}/\alpha$ | $\Delta_{\text{2lvl}}/\alpha$ | $\Gamma/2\omega_{n,n'}^R$ |
|---|---|---|---|---|---|---|---|---|
| 1 | 0 | 3 | 0 | 0.034966 | 2000 | 1.5 | 1.5 | 0 |
| 2 | 0 | 3 | -0.025 | 0.029492 | 2000 | 1.274905 | 1.274905 | 0.015 |
| 3 | 0 | 4 | 0 | 0.099034 | 2000 | 2.0 | 2.0 | 0 |
| 4 | 0 | 4 | -0.025 | 0.075692 | 2000 | 1.599393 | 1.599395 | 1.1 |
| 5 | 0 | 5 | 0 | 0.202931 | 1000 | 2.5 | 2.5 | 0 |
| 6 | 0 | 5 | -0.025 | 0.138884 | 100 | 1.872625 | 1.872634 | 1.3 |
| 7 | 0 | 5 | 0.025 | 0.261639 | 1000 | 3.125676 | 3.125674 | 1.1 |

Table 1: The parameters chosen for the simulations of multi-photon transitions between Fock states $|n\rangle$, $|n'\rangle$ with full model (3) and two-level model (10). In all cases, the multi-photon transition period is close to $3 \times 10^4 \alpha^{-1}$.

For each transition, we chose $g_0$ to get the multi-photon Rabi oscillation period $\alpha T_R \approx 3 \times 10^4$. We considered the same relative amplitude fluctuations $\eta = 0.1$ for all transitions. For transitions in the over-Kerr model by 4 and 5 quanta, we chose the correlation time $\tau$ to satisfy $\Gamma/2\omega_{n,n'}^R > 1$ (overdamped regime). For the Kerr model, we chose similar correlation times.

For transition by 3 quanta, the condition $\Gamma/2\omega_{n,n'}^R > 1$ can be satisfied only for very large correlation times, according to the analysis in Section 3 (see Fig. 4). So, we choose $\alpha\tau = 2000$ as for transitions by 4 quanta both for the Kerr and over-Kerr models.

In Fig. 5 – 6, we demonstrate the results of the numerical simulations together with analytical calculations. For all of the cases, we obtained quite a good agreement between the analytical predictions and numerical results.

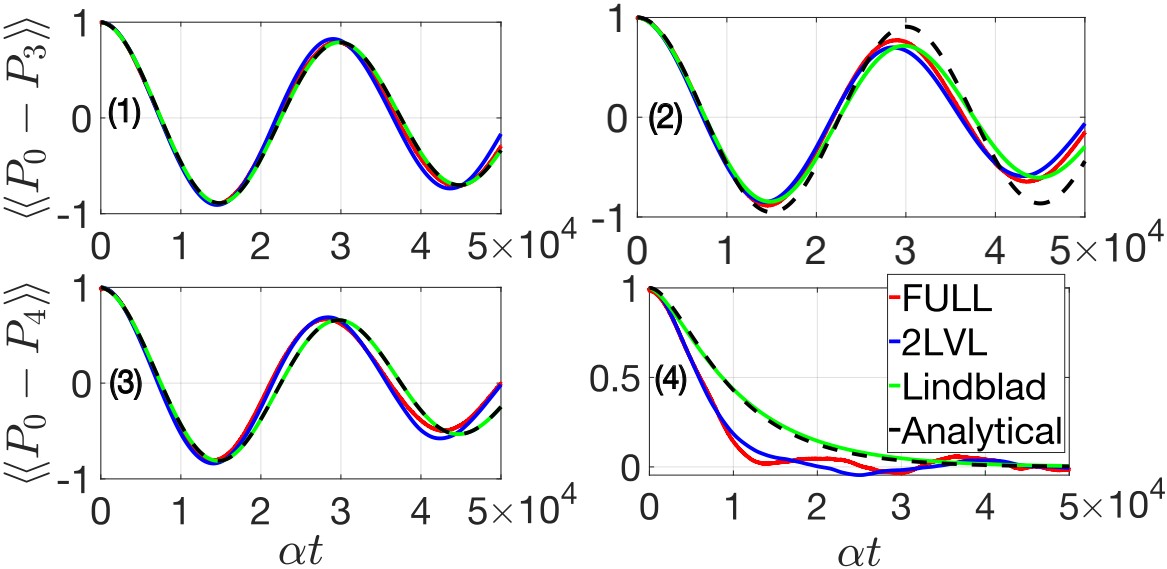

Figure 5: Population inversion as a function of time for transitions by 3 and 4 quanta (panels (1)–(4) correspond to rows 1–4 of Table 1). For all of the figures, the red (blue) line corresponds to the noise-averaged simulations ($\sim 10^3$) of TDSE for the full (3) (two-level (10)) model. Green lines correspond to the numerical solution of the master equation (15). Black dashed line corresponds to the analytical solutions (16) and (18). The cut-off for the Hilbert space is 11 quanta. The error bar for all of the figures is below 5 %.

For transitions by 4 and 5 quanta, our results demonstrate the significantly different influence of the amplitude noise for the Kerr and over-Kerr models. As one can see, in the over-Kerr model, the multi-photon oscillations become suppressed, whereas they remain pronounced in the Kerr one for similar noise parameters. Meanwhile, for the transition by 3 quanta, multi-photon oscillations survive for both Kerr and over-Kerr models, as $\Gamma/2\omega_{n,n'}^R < 1$ for the chosen set of parameters.

In our simulations, we see a good coincidence between the results of the full model and the two-level effective one. The possible discrepancy between these models can be caused by noise-induced transitions to the non-resonant quasienergy levels and the fact that the true eigenstates are not the Fock states but dressed states. Also, we see a discrepancy between these results and the master equation solution. The latter can be attributed to the weak-noise approximation (see Eq. (37)) which causes a noticeable error at moderate $g_0$, and $\eta$.

## 5 Conclusion

In this work, we studied multi-photon transitions in the model of quantum nonlinear oscillator in the presence of high-order nonlinearities. We examined the robustness of multi-photon transitions to the driving field fluctuations.

For that, we utilized a two-level effective model for two resonant oscillator levels which is valid for relatively large noise correlation times.

Using analytical and numerical results for the two-level effective model and full TDSE simulations, we demonstrated the robustness of the multi-photon transitions in the Kerr

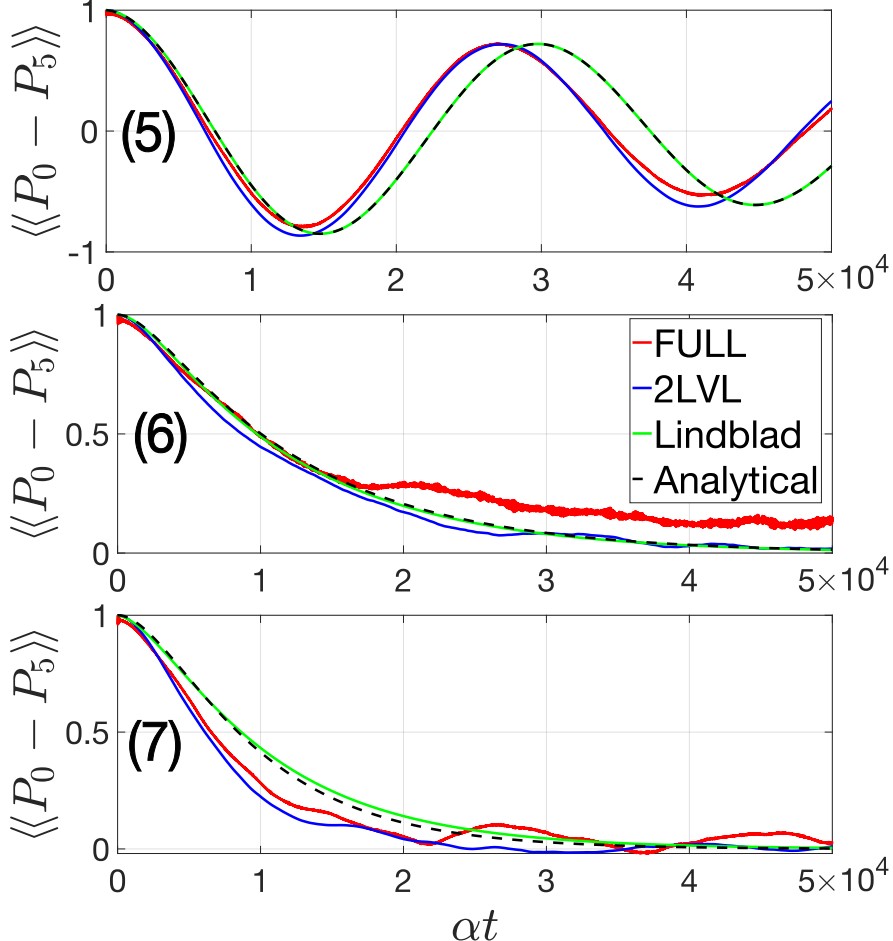

Figure 6:   Population inversion as a function of time for transitions by 5 quanta (panels (5)–(7) correspond to rows 5–7 of Table 1). For simulation details, see the caption to Fig. 5.

oscillator to the amplitude fluctuations.

In contrast, for the oscillator with higher-order nonlinearity (over-Kerr), we showed that the dominant contribution to the decay of the multi-photon Rabi oscillations comes from the oscillator quasienergy level shifts induced by the driving field. Using the master equation for the two-level model, we find analytical expressions for the decay rate of multi-photon oscillations. For the Kerr oscillator, quasienergy shifts vanish due to the equality of perturbation theory corrections to the quasienergy levels, which leads to a considerably smaller decay rate.

Also, we found out that the transitions by different numbers of quanta in the presence of high-order nonlinearities have different sensitivity to the driving field fluctuations.

Regarding potential experimental observations, we have observed that, in principle, multi-photon Rabi oscillations can be observed in a model involving two cold-trapped ions under the influence of an external driving field. Achieving this requires precise tuning of physical parameters, such as the driving field's frequency. However, based on our estimations, we believe these parameters are realistic from an experimental standpoint. The precise calculations fall beyond the scope of this work.

Our findings pose limitations on the driving field noise level necessary for experimental observation of multi-photon Rabi oscillations, which can have applications for the manipulation of the state of the quantum oscillators.

## Acknowledgements

The authors thank Nikolay Gippius for useful discussions.

**Funding information**    The work of B. Y. N. was supported by the Russian Roadmap for Quantum computing (Contract No. 868-1.3-15/15-2021 dated October 5, 2021).

## A    The difference of the quasienergy corrections

In this Appendix, we demonstrate that the second order quasienergy corrections are always different for the over-Kerr model at resonant detuning. In second-order perturbation theory, the quasienergy levels of the oscillator (3) equal $\epsilon_n(g) = \epsilon_n^{(0)} + \epsilon_n^{(2)} g^2$, where

$$
\epsilon_n^{(0)} = -\Delta n + \frac{\alpha}{2} n^2 + \kappa n^3,
$$
$$
\epsilon_n^{(2)} = \frac{n}{\epsilon_n^{(0)} - \epsilon_{n-1}^{(0)}} + \frac{n+1}{\epsilon_n^{(0)} - \epsilon_{n+1}^{(0)}}.
\tag{19}
$$

As shown in Section 3, the sensitivity of multi-photon Rabi oscillations to amplitude noise depends on the value of the difference $\epsilon_n^{(2)} - \epsilon_{n'}^{(2)}$ between second-order corrections at $\Delta = \Delta_{\mathrm{res}}^{(0)}$. We obtained an expression for the corrections difference in the first order by $\kappa$

$$
\epsilon_n^{(2)} - \epsilon_{n'}^{(2)} = \frac{4(n-n')(n+n'+1)}{\alpha^2 \left((n-n')^2 - 1\right)} \kappa + o(\kappa).
\tag{20}
$$

According to Eq. (20), the correction vanishes at $\kappa = 0$ and is non-zero at $\kappa \neq 0$. Also, see the plots of for the values of $\epsilon_n^{(2)} - \epsilon_{n'}^{(2)}$ in Fig. 7 obtained directly from Eq. (19) without expansion in $\kappa$.

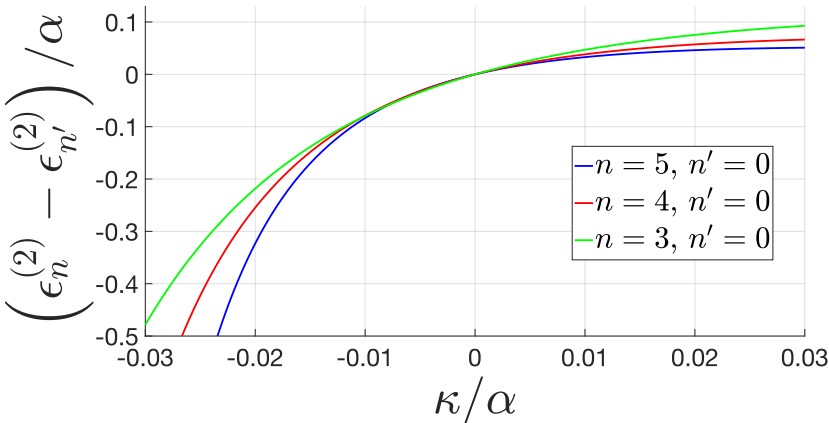

Figure 7: Second order quasienergy corrections difference as a function of $\kappa$ at resonant detuning. Here curves corresponds to the resonance between the states: $n = 5$ and $n' = 0$, $n = 4$ and $n' = 0$, $n = 3$ and $n' = 0$.

## B  Coloured noise

In this Appendix, we derive the condition of applicability of the two-level effective Hamiltonian (10) for the description of the system near resonance between levels $n$ and $n'$.

Let us consider the Hamiltonian in form

$$H(t) = \sum_n \epsilon_n \, |n\rangle\langle n| + \sum_{j=1}^{2} \sum_{n,n'} \xi_j(t) \, (V_j)_{n,n'} \, |n\rangle\langle n'| , \tag{21}$$

where $|n\rangle$ are the eigenstates of the stationary Schroedinger equation (not Fock states) with quasienergies $\epsilon_n$. Also, $V_1 = a + a^\dagger$, and $V_2 = a^\dagger a$, see Eq. (3). We assume the noise sources $\xi_i(t)$ have the correlation functions

$$\langle \xi_i(t)\xi_j(t')\rangle = C_i((t - t')/\tau_i)\delta_{ij}, \quad i,j = 1,2. \tag{22}$$

To find the transition rate to non-resonant levels due to noise, let us use the first-order perturbation theory, where noise term is considered as a perturbation. Let us expand the evolution operator up to the first order

$$U = \mathbb{1} - i \int_0^T H_{\mathrm{I}}(t)dt + \dots, \tag{23}$$

where the interaction picture Hamiltonian reads

$$H_{\mathrm{I}}(t) = \sum_{j=1}^{2} \sum_{n,n'} \xi_j(t) \, (V_j)_{n,n'} \, e^{i(\epsilon_n - \epsilon_{n'})t} \, |n\rangle\langle n'| . \tag{24}$$

Let us introduce for brevity of the notations

$$\delta\epsilon_{n,n'} \equiv \epsilon_n - \epsilon_{n'}. \tag{25}$$

Thus, the probability of the transition between the states $n'$ and $k$ in the first-order of perturbation theory over the time interval $T$ reads

$$P_{n' \to k} = \left| \langle k | U | n' \rangle \right|^2 \approx \int\limits_{[0,T]^2} \left| \sum_{j=1}^{2} \xi_j(t) (V_j)_{k,n'} \right|^2 e^{i\delta\epsilon_{k,n'}(t-t')} dt dt'. \tag{26}$$

Averaging (26) over the noise, one can find

$$\langle\langle P_{n' \to k} \rangle\rangle \approx \sum_{j=1}^{2} \int\limits_{[0,T]^2} \left| (V_j)_{k,n'} \right|^2 C_j((t-t')/\tau_j) e^{i\delta\epsilon_{k,n'}(t-t')} dt dt'. \tag{27}$$

In our consideration, $T$ has the order of multi-photon transition period, and we assume that noise correlation times are much smaller than $T$. Thus, the double integral in (27) can be approximated by

$$\langle\langle P_{n' \to k} \rangle\rangle \approx T \sum_{j=1}^{2} \tau_j \left| (V_j)_{k,n'} \right|^2 \mathcal{F}[C_j]\left(-\tau_j \delta\epsilon_{k,n'}\right) + o(T), \tag{28}$$

where $\mathcal{F}[C](z)$ is a Fourier transform of the correlation function

$$\mathcal{F}[C](z) = \int\limits_{-\infty}^{+\infty} C(s) e^{-izs} ds. \tag{29}$$

For the exponentially correlated amplitude noise (without frequency noise)

$$C(x) = \sigma^2 \exp\left(-|x|\right), \quad \mathcal{F}[C](z) = \frac{2\sigma^2}{1+z^2}, \tag{30}$$

the transition probability reads

$$\langle\langle P_{n' \to k} \rangle\rangle \approx \frac{Q \left| V_{k,n'} \right|^2}{1 + \delta\epsilon_{k,n'}^2 \tau^2} T, \tag{31}$$

where $Q = 2\tau\sigma^2$. For the applicability of the two-level approximation (10), we require that

$$\langle\langle P_{n' \to k} \rangle\rangle \ll 1 \tag{32}$$

for $T$ comparable to the period of multi-photon transitions. Neglecting the unit in the denominator in Eq. (31) and assuming $T \sim T_R = 2\pi/\omega_{n,n'}^R$ (with $n$, and $n'$ be the resonant levels), we get

$$\tau \gg \frac{4\pi\sigma^2 \left| V_{k,n'} \right|^2}{\left(\delta\epsilon_{k,n'}\right)^2 \omega_{n,n'}^R} \sim \frac{2\sigma^2 T_R}{\alpha^2}. \tag{33}$$

In the last estimate, we used that $\delta\epsilon_{k,n'} \sim \alpha$ for the non-resonant level $k$, and $\left| V_{k,n'} \right|^2 \sim 1$ for transitions between low-lying oscillator levels by small number of quanta. From Eq. (33) it is clear the correlation time should be large enough for the validity of the two-level approximation. Still, for small noise amplitude $\sigma$, it can be much smaller than the period of multi-photon transitions.

## C   Exact solution of the master equation for the two-level effective model

In this Appendix, we discuss the master equation approach to treat the two-level effective system with classical noise Eq. (13) – (14). We obtain the solutions (16) and (18) for the master equation. We also demonstrate that the presence of the frequency noise leads to the destruction of the Rabi oscillations.

Let us assume the Hamiltonian can be written in the following form

$$H(t) = H_0 + \sum_{j=1}^{2} \xi_j(t) V_j(t), \tag{34}$$

where $\xi_j(t)$ — coloured Gaussian real noises with zero mean and correlation functions $\langle \xi_i(t) \xi_j(t') \rangle = C_i((t-t')/\tau_i)\delta_{ij}$. One can find [26, 25] that the master equation for the density matrix reads (for the case when two processes with $\xi_1(t)$, and $\xi_2(t)$ are independent)

$$\dot{\rho} = -i[H_0, \rho] - \sum_{j=1}^{2} \left[ V_j(t), \int_0^t dt' C_j\left((t-t')/\tau_j\right) \left[V_j(t'), \rho(t')\right] \right]. \tag{35}$$

For white noise, $C(x) = Q\delta(x)$, Eq. (35) reduces to the form of GKSL equation [27, 28]:

$$\dot{\rho} = -i[H_0, \rho] + \sum_{j=1}^{2} \frac{Q_j}{2} \left( 2V_j \rho V_j - \{V_j^2, \rho\} \right). \tag{36}$$

Let us apply this result to the effective Hamiltonian (10). We assume that the classical noise amplitude $\xi_1(t)$ is small, which allows to perform the expansion

$$g(t)^k = (g_0 + \xi_1(t))^k \approx g_0^k + k g_0^{k-1} \xi_1(t). \tag{37}$$

Under this assumption, the effective Hamiltonian (10) reduces to the form (34) with

$$
H_0 = \begin{bmatrix} \epsilon_{n'}^{(0)} + \epsilon_{n'}^{(2)} g_0^2 & \omega_{n,n'} g_0^{n-n'} \\ \omega_{n,n'} g_0^{n-n'} & \epsilon_n^{(0)} + \epsilon_n^{(2)} g_0^2 \end{bmatrix} \equiv \begin{bmatrix} H_{11} & H_{12} \\ H_{12} & H_{22} \end{bmatrix},
$$
$$
V_1 = \begin{bmatrix} 2\epsilon_{n'}^{(2)} g_0 & \omega_{n,n'}(n-n') g_0^{n-n'-1} \\ \omega_{n,n'}(n-n') g_0^{n-n'-1} & 2\epsilon_n^{(2)} g_0 \end{bmatrix} \equiv \begin{bmatrix} V_{11} & V_{12} \\ V_{12} & V_{22} \end{bmatrix}, \tag{38}
$$
$$
V_2 = \begin{bmatrix} -n' & 0 \\ 0 & -n \end{bmatrix}.
$$

The evolution for the noise-averaged density matrix for the system (10) is described by (36).

It is useful to expand the density matrix in the basis of Pauli matrices [29]

$$\rho = \frac{1}{2} + \rho_x \sigma_x + \rho_y \sigma_y + \rho_z \sigma_z. \tag{39}$$

Plugging (39) into (36), one can obtain the Cauchy problem with the initial conditions: $\rho_z(0) = 1/2$, and $\rho_x(0) = \rho_y(0) = 0$ for the following system

$$\dot{\rho}_x = -\frac{1}{2}\left[Q_1\left(V_{11} - V_{22}\right)^2 + Q_2\left(n - n'\right)^2\right]\rho_x(t) + \left(H_{22} - H_{11}\right)\rho_y(t) + Q_1 V_{12}\left(V_{11} - V_{22}\right)\rho_z(t),$$

$$\dot{\rho}_y = \left(H_{11} - H_{22}\right)\rho_x(t) - \frac{1}{2}\left[4Q_1 V_{12}^2 + \left(V_{11} - V_{22}\right)^2 + Q_2\left(n - n'\right)^2\right]\rho_y(t) - 2H_{12}\rho_z(t),$$

$$\dot{\rho}_z = Q_1 V_{12}\left(V_{11} - V_{22}\right)\rho_x(t) + 2H_{12}\rho_y(t) - 2Q_1 V_{12}^2\rho_z(t).$$

$$(40)$$

We consider the solution of (40) in three simplifying cases:

1. the over-Kerr oscillator with both frequency and amplitude noise,

2. the Kerr oscillator with frequency noise only,

3. the Kerr oscillator with amplitude noise only.

For all cases, we assume resonance condition $(H_{11} = H_{22})$. For the over-Kerr oscillator, we use the condition $V_{12} \ll \min(V_{11}, V_{22})$ as $V_{12} \propto g_0^{n-n'-1}$, and $V_{11}, V_{22} \sim g_0^2$. Neglecting $V_{12}$, we find the solution

$$\rho_z(t) = \frac{1}{2}e^{-\Gamma t}\left[\cosh\left(t\sqrt{\mathcal{D}}\right) + \frac{\Gamma}{\sqrt{\mathcal{D}}}\sinh\left(t\sqrt{\mathcal{D}}\right)\right], \tag{41}$$

where

$$\Gamma = Q_1\left(V_{11} - V_{22}\right)^2/4 + Q_2\left(n - n'\right)^2/4 = Q_1 g_0^2\left(\epsilon_n^{(2)} - \epsilon_{n'}^{(2)}\right)^2 + Q_2\left(n - n'\right)^2/4,$$
$$\mathcal{D} = \Gamma^2 - 4H_{12}^2. \tag{42}$$

For the Kerr oscillator with frequency noise only, the solution takes exactly the same form as (41), (42) with $Q_1 = 0$, $\kappa = 0$.

For the Kerr oscillator with purely amplitude noise, the solution reads

$$\rho_z(t) = \frac{1}{2}e^{-2Q_1 t V_{12}^2}\cos\left(2H_{12}t\right). \tag{43}$$

As one can see from Eqs. (41) – (43) the contribution of the frequency noise to the decay rate of both Kerr and over-Kerr oscillator is typically much larger than the contribution of the amplitude noise. Thus, the presence of the frequency noise leads to the destruction of multi-photon Rabi oscillations in both Kerr and over-Kerr models, and we do not consider it in our analysis in the main text. In contrast, pure amplitude noise leads to the substantially different decay rates (41) and (43) for the Kerr and over-Kerr cases.

Moreover, for the Kerr model, the noise-averaged population inversion can be calculated for arbitrary Gaussian correlated amplitude noise. The Hamiltonian and the evolution operator reads

$$H(t) = \left(H_{12} + \xi(t)V_{12}\right)\sigma_x + \text{const}, \quad U(t) = \exp\left[-i\sigma_x\int\limits_0^t\left(H_{12} + \xi(t')V_{12}\right)dt'\right]. \tag{44}$$

One can find for the noise-averaged population inversion

$$\langle\!\langle\sigma_z(t)\rangle\!\rangle = \langle\!\langle U^\dagger\sigma_z U\rangle\!\rangle = \langle\!\langle\cos 2\phi(t)\rangle\!\rangle = \text{Re}\langle\!\langle e^{2i\phi(t)}\rangle\!\rangle, \tag{45}$$

where $\phi(t) = \int_0^t \left( H_{12} + \xi(t')V_{12} \right) dt'$. With the help of the properties of Gaussian noise, one can find

$$\langle\langle \sigma_z(t) \rangle\rangle = \exp\left[ -2V_{12}^2 \int\limits_{[0,t]^2} \langle \xi(t')\xi(t'') \rangle dt'dt'' \right] \cos\left( 2H_{12}t \right). \tag{46}$$

# D Lower bound for the period of the multi-photon Rabi oscillations

In this Appendix, we discuss the lower bound for the period of multi-photon Rabi oscillations necessary to reach overdamped regime for the over-Kerr model.

According to Eq. (17), the over-Kerr oscillator in overdamped regime when $\Gamma > 2\omega_{n,n'}^R$. One can rewrite this condition as follows

$$\frac{\eta^2\tau}{\omega_{n,n'}g_0^{n-n'-4}} \left( \epsilon_n^{(2)} - \epsilon_{n'}^{(2)} \right)^2 > 1 \tag{47}$$

However, the white-noise approximation for the two-level effective model is valid only for $\tau \ll 2\pi/\omega_{n,n'}^R$. We would like to estimate the time $\tilde{T}$ when the system reaches the overdamped regime. Combining these two conditions, from Eq. (47), one can write

$$g_0 \ll \left( \frac{\sqrt{2\pi}\eta \left| \epsilon_n^{(2)} - \epsilon_{n'}^{(2)} \right|}{\omega_{n,n'}} \right)^{\frac{1}{n-n'-2}}. \tag{48}$$

Thus, for the period of Rabi oscillations, one can find

$$T_R = \frac{2\pi}{\omega_{n,n'}^R} \gg \frac{2\pi}{\omega_{n,n'}} \left( \frac{\omega_{n,n'}}{\sqrt{2\pi}\eta \left| \epsilon_n^{(2)} - \epsilon_{n'}^{(2)} \right|} \right)^{\frac{n-n'}{n-n'-2}} \equiv \tilde{T}. \tag{49}$$

Since the perturbation theory corrections difference is proportional to $\kappa$ (see Appendix A), one can find that the value $\tilde{T}$ scales as $|\kappa|^{-1-2/(n-n'-2)}$.

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
