# Peer review of "Multi-photon Rabi oscillations in the presence of the classical noise in a quantum nonlinear oscillator"

_SciPost Physics_

## Round 1 · Referee Report · Anonymous (Referee 1) · 2025-4-1

Strengths

  1. The authors provide a comprehensive analysis distinguishing between Kerr and super-Kerr (also referred to as over-Kerr in this work) anharmonicities, with a focus on their respective sensitivities to fluctuations in the driving field. The key finding is that multiphoton transitions induced by Kerr nonlinearity are robust to such fluctuations, attributed to a symmetry in the perturbative corrections to the oscillator quasienergies. This symmetry, however, is absent in over-Kerr systems.

  2. Their findings establish constraints on the allowable noise levels in the driving field for the experimental realization of multiphoton Rabi oscillations—an important consideration for quantum state manipulation in nonlinear oscillators.

  3. The methodology used is rigorous and well-founded.

Weaknesses

  1. The authors should discuss/highlight a few concrete physical examples, supported by empirical parameter values, to highlight the regimes in which nonlinear systems are expected to follow the different trends theoretically uncovered in their work.

  2. While the results/predictions are interesting, the exact relevance of their work concerning any potential future developments of the topic has to be addressed more convincingly. For instance, what kind of systems feature a large over-Kerr type anharmonicity?

Report

While the manuscript is publishable subject to minor revisions, I believe the correct categorization of the criterion fulfilled by this work should be:

Opens a new pathway in an existing or a new research direction, with clear potential for multi-pronged follow-up work.

Having said that, a stronger justification for this criterion is warranted.

Requested changes

The authors should address the following:

  1. In the Introduction section, 2nd paragraph, what is meant by the ultra-quantum regime? Please clarify.

  2. The authors claim that the Hamiltonian in Eq. (1) corresponds to a resonant driving field. What is the resonance condition?

  3. In Eq. (6), why do only even powers of $g$ appear? A brief clarification is warranted.

  4. In Section 3, the two-level approximation is premised on the assumption of a large correlation time in the input noise source. However, later, the authors also investigate the case of delta-correlated noise. They should briefly summarize/enumerate the conditions under which their model remains valid in either scenario.

  5. In the concluding section, scant details are provided regarding the future outlook. Please furnish more details.

Recommendation

Ask for minor revision

---

## Editorial Decision

awaiting_resubmission